# Voices of surgical wards nurses on barriers hindering acute post-operative pain management at Tshwane municipality, South Africa

**Nnene Melia Makou**[1]\*, **Melitah Molatelo Rasweswe**[2], **Ramadimetja Shirley Mooa**[1]

**1** Department of Nursing Science, Faculty of Health Sciences, University of Pretoria, Gauteng Province, South Africa, **2** Department of Nursing Science, Faculty of Health Sciences, University of Limpopo, Limpopo Province, South Africa

\* melia.makou@gmail.com

## Abstract

### Introduction and background

Acute pain is expected following a surgery, but it is often inadequately managed by health care providers. However, little is known about the barriers that hinder acute post-operative pain management among surgical wards nurses.

### Objective

Uncovering barriers that hinder the surgical wards nurses to manage acute post-operative pain at the selected public hospital in Tshwane municipality, Gauteng Province, South Africa.

### Methods

This study utilized a qualitative explorative, descriptive, and contextual research design. Individual semi-structured interviews were conducted from a purposive sampling of 13 professional nurses. Data collected were audio recorded and transcribed verbatim by the first author. Data were analysed using thematic data analysis, which led to the emergence of themes and sub-themes. An independent co-coder assisted with data analysis.

### Findings

The barriers described by the participants include: 1) Organisational/ management related barriers; 2) Personnel related barriers, which were discussed as shortage of nurses, inadequate skill competency to manage acute post-operative pain, and interprofessional communication; and 3) Patient related barriers.

### Conclusion

This paper comes to the conclusion that, due to a number of stated barriers or obstacles, the nurses employed in the surgical wards of the selected hospital in South Africa are not

**Data Availability Statement:** All relevant data are within the manuscript.

**Funding:** The author(s) received no specific funding for this work.

**Competing interests:** The authors have declared
that no competing interests exist.

adequately managing the acute post-operative pain. As a result, hospital management must devise practical solutions to the stated obstacles.

## Introduction

The acute post-operative pain management, is a comprehensive, deliberate action taken by health care providers to relieve patients pain following surgery, using both pharmacological and non-pharmacological interventions [1]. It is crucial to effectively control post-operative pain, since several studies have confirmed that acute post-operative pain is prevalent and inevitable in all patients that undergo a surgical procedure [2]. While, it is acknowledged that post-operative pain is a common consequence of surgery, evidence suggests that the levels of post-operative pain experienced by patients are unnecessarily high. Reports state that acute discomfort affects approximately 75% of people undergoing surgical procedures, and between 20 to 80 percent of patients feel pain or acute discomfort after surgery [3].

Africa recognizes the global belief that it is critical to treat acute post-operative pain, but many patients still suffer from post-operative pain [4]. This, is largely detailed to be associated with barriers such as, unawareness, inadequate knowledge, work overload and lack of time, among health care providers including nurses. A Tanzanian study reported that, 70% of patients experienced moderate to severe postoperative pain, and out of that, 14.2% had numerical rating scale (NRS 0–10) ratings greater than 7 [5]. A South African study reported that, 62% of patients experience moderate or severe post-operative pain [6]. Although there is no accurate data on the incidence of acute post-operative pain in Tshwane municipality hospitals, the findings from literature suggest that there may be problems.

Pain itself is so complex, and involves the central and peripheral nerve systems. that may have an effect on a patient's physical, social and psychological health [7]. There is a consensus in the literature, that pain is subjective rather than objective in nature [8–10]. Hence, it is becoming more widely acknowledged that pain in general is an individualized experience that is influenced by a variety of factors, including mood, life events, fear, anxiety, and anticipation. Regarding, acute post-operative pain, there is a consensus that it affects patients on both physiological, psychological and social level [11–13].

Acute post-operative pain can also lead to a number of surgical complications, including but not limited to a general deterioration in quality of life, a higher risk of thromboembolic events and respiratory problems, and longer hospital stay [14]. A long-lasting pain, without relief, will eventually turn into chronic pain. Chronic pain results in a whole new set of issues that affect individuals, their families, communities, and the healthcare system [14, 15]. Good postoperative pain management can reduce the emotional and physical strain that patients endure following surgery, which in turn lowers the risk of problems after surgery and speeds up the body's healing process. Therefore, in order for patients to be pain free, they should be appropriately managed by the health care providers, especially nurses who spend a lot of time with the patients. However, postoperative pain management reported to be a significant problem and challenge in many settings, particularly in low- and middle-income countries [7].

South Africa is not an exception; a study conducted in the Western Cape province indicates that there is a lack of patients' post-operative assessment, as well as implementation of management guidelines [9]. The main obstacles to post-operative pain care in Africa have been determined to be insufficient assessment, staff ignorance, patients' misconceptions, a lack of resources, and medication limitations [16–18]. While reducing the intensity of acute post-

operative pain is the primary duty of all health care providers, nurses are in the front line and must possess strong understanding in this area. Thus, the nurses' knowledge and attitude have a valuable effect in keeping the patients free from the pain following surgery.

In South Africa, nurses role regarding post-operative pain include, effectively assessing, monitoring pain levels, communicating, managing through administration of pharmacological and non-pharmacological means, as well as documenting findings and interventions [19]. Although nurses act as key managers in relieving patients post-operative pain, some studies noted that the extent of "good knowledge" of post-operative pain management among nurses is low, in Northwest Ethiopia was 66.9% [20], and 61.1% in northern Ghana [21]. Additionally, little is known about the barriers that hinder them to manage acute post-operative pain. This gap was closed by exploring the barriers that hinder the surgical wards nurses to manage acute post-operative pain at the selected public hospital in Tshwane municipality, Gauteng Province, South Africa.

## Methods

### Study design and setting

A qualitative explorative, descriptive and contextual research design was followed, to explore the barriers that hinder surgical ward nurses to manage acute post-operative pain. The study was conducted at an academic public hospital, which is situated in the city of Tshwane Metropolitan Municipality in the Gauteng province, South Africa. This public hospital provides care to a diverse population across Gauteng province, and neighbouring provinces such as North West and Limpopo. The hospital has 832 beds, housed in twenty-eight wards. Out of all the wards, three are adult surgical wards. The adult surgical wards admit both males and females who underwent major general, urology and orthopaedic surgery. Each surgical ward caters for approximately 40 patients. Eight to nine patients are admitted in each ward post surgery daily. Different categories of nurses are placed in surgical wards to provide nursing care twenty-four hours. In each surgical ward there are six professional nurses, who work shifts to meet the demands of the patients.

### Population and sampling

The target population were both male and female professional nurses, registered by a nursing regulatory authority South African Nursing Council, and working at the three surgical wards of the selected public hospital for more than twelve months. Other categories of nurses and student nurses were excluded from taking part. In addition, participants in this study were not registered professional nurses employed on a contract or temporary basis in the public hospital's surgical wards. In order to identify the most typical participant qualities, purposeful non-probability sampling was used, based on the judgement of the researchers and their involvement in the care of post-operative patient. A total number of thirteen were interviewed, and the number depended on data saturation.

### Data collection and organization

Participants were recruited during staff meetings, in order to minimize interference with their work schedule. Appointments were made with the participants and agreed on the date and time for interview. All the participants were interviewed outside their regular working hours, such as tea time, lunch and after work at the ward conference room, which is a natural setting for them. A written informed consent was obtained prior to participation. Data was collected through individual semi-structured interviews for two months, between March 2023 and April

2023. Data were collected on various days until saturation was reached at participant 10, but three more were interviewed to verify that there is no more new information emerging. The interviews were conducted in English since all the participants were able to read and write in English. A focal question during interviews was "*what are the barriers that hinder surgical ward nurses to manage acute post-operative pain in your setting*?". Probing questions were asked, depending on the response of an individual participant. The interviews lasted between thirty and forty-five minutes. An audio recorder was used to capture the conversation between the interviewer and interviewee. Additionally, the non-verbal language was noted to verify data during analysis.

## Data analysis

The six steps of reflexive thematic data analysis as described by [22] were followed. In the first step, the first author listened to each audio recorded interview and transcribed the data verbatim. Nonverbal cues from participants were included in the data transcription. In order to properly comprehend and have a sense of the participants' ideas, she read and re-read transcripts several times. Step 2 involved identifying the informative elements of the data items, and organizes them into meaningful groups or codes. All codes were transferred to the spreadsheet. And individual participant's information was labelled accordingly. Important patterns were identified from codes and highlighted with different colours. In step 3, codes were reviewed and an analysis was conducted to determine how various codes may be joined based on related meanings to create themes or sub-themes. This was discussed and agreed with the co-authors who had not taken part in the coding process. Step 4, involved setting aside and discarding themes that did not pertain to the study questions. In order to aid in the meaningful interpretation of the data, certain codes and themes were revised or eliminated. The relevant themes were divided into broader categories, after which the themes were separated accordingly. Step 5 involved, naming themes in order to organise the barriers that hinder post-operative pain management by the surgical ward nurses. The final step entailed creating the final report, which included a brief and simple description of the themes.

## Ethical consideration

Prior to conducting the study, ethical clearance certificate was received from the University of Pretoria Health Sciences Research Ethics Committee (Ethics reference number: 717/2022). Permission to conduct the study in the selected hospital was obtained from the Gauteng Department of Health (NHRD Ref Number: GP_202302–009), and hospital's management. The participants' confidentiality and anonymity were maintained throughout the study, instead of using the participants, numbers were assigned in relation to gender and ward, for example P10MU.

## Results

### The socio-demographic characteristics of the participants

Female participants were the largest group (12) compared to (1) male participant. All the participants were black professional nurses, their ages ranged between 30 and 65 years. Years of working experience in adult surgical wards at the selected hospital ranged between one year and 20 years. Three participants were from general surgical ward, three from urology, and seven from orthopaedic. The socio-demographic characteristics of the participants are presented, in Table 1.

**Table 1. The socio-demographic characteristics of participants.**

| VARIABLE | CHARACTERISTICS | FREQUENCY |
|---|---|---|
| Gender | Male(s) | 1 |
| | Females | 12 |
| Age | 30 to 45 years | 6 |
| | 46 to 65 years | 7 |
| Nationality | Black | 13 |
| | White | 0 |
| Working experience in a surgical ward | 2 to 10 years | 6 |
| | 11 to 20 years | 7 |
| Name of surgical ward | General surgical ward | 3 |
| | Urology surgical ward | 3 |
| | Orthopaedic surgical ward | 7 |

## Barriers hindering acute post-operative pain management

The barriers hindering acute post-operative pain management were grouped into 3 themes and 3 subthemes. The presentation of results supported by the participants quotes.

**Theme 1. Organisational/management related barriers.** The organisational/management related barriers were discussed in relation to increased workload or delegation. The participants argued that workload in the surgical ward, has an impact on the quality of acute post-operative pain. They expressed concerns regarding an increased workload, which leads to inadequate quality of nursing care to the patients. Nurse from urology ward mentioned that: *"Number one, it can be a busy ward. When the ward is busy, we as nurses can turn not to care like giving our patients rightful care because the ward is too busy, for example, I am admitting the patient that side and I needed to fetch other patients from theatre. And it is not only one patient, it is more, for example in the ward, every day we do operations to patients. Here, I have to attend to pre-ops patients and they want the patient as like now. They want to take the patient to theatre when the theatre calls come to fetch the patient. I go fetch the patient from theatre, I need to attend to the patient, but I am busy. . . raising both hands up. . . so we do manage post-operative pain routinely, but is not how it should be"* (P2FU).

A general surgical ward nurse said: *"In this ward we can operate ten patients a day, so it becomes difficult to give patients full attention to assess and manage patients' post-operative pain. It is so busy that when you are still attending to one patient, operating theatre call you to fetch the other patient, and in between you might resuscitate a patient, and you need to do other nursing duties, workload is really crazy in here"* (P3FG).

The other participant mentioned that: *". . . Like for instead I am busy when the patient complains of pain. I request one of the nurses to give the patient pain medication and you will find that the nurse is busy attending to other patients. Then you find the timeframe. . . that we give the patient medication after he had complained a lot about pain"* (P11FO).

The participants stated that they are not coping with the increased workload in the surgical ward to can manage acute post-operative pain adequately. *"I am not coping because I am doing. . ., many activities at a go alone, and I am only one sister doing vital signs, fetching patients from theatre, I give medication and I admit the patients. We are admitting every day in surgical ward and we are operating every day. It is very difficult for us, we are becoming dangerous to post-operative patients slowly but surely, opening both eyes widely"* (P5FO).

The other participant revealed that increased workload lead to poor quality of care regarding management of acute post-operative pain. *"Actually, we are not coping. We are working*

*because we come to work. The quality of nursing care we give to our patients is zero; that is why we are having a high rate of complaints in the unit due to poor nursing care. We like to provide high quality care, but it is highly impossible due to overload, it's too much for all of us actually"* (P1FO).

Participants also revealed that, due to increased workload at times they do not follow the scope of practice as stipulated by the South African Nursing Council. One participant said:

*"Increased workload drive us, when it comes to delegation,. . . . We do not even delegate according to our scope of practice. We just allocate, we just work looking down, at times we only when patient scream because of pain, we then run around in the ward, trying to manage"* (P7FG).

The participants stated that increased workload and not working according to the scope of practice lead to burnout, which in turn promotes absenteeism in the surgical wards. *"Burnout, I am talking about tiredness, caused by increased workload, and it promotes absenteeism as well because when you are tired you do not feel like going to work, I mean you're tired, pushing hair back, and frowning"* (P2FU).

## Personnel related barriers: Shortage of nurses, inadequate skill competency and interprofessional communication

The nurse related barriers was identified as a serious challenge in the surgical wards of the selected hospital, regarding management of acute post-operative pain. This barrier was mostly described under shortage of nurses, inadequate competency to manage post-operative pain and interprofessional communication.

**Shortage of nurses.** It was identified that there is a high shortage of nurses, which makes it difficult for the available nurses to manage the high demands found in the surgical wards. Participants indicated that they were not attending to patients on time due to a shortage of nurses. This has a negative effect on the rendering quality nursing care, such as not assessing and managing post-operative pain, especially, failing to administer pain medication to the patients.

The participants said: *". . . I will say sometimes is shortage of staff. So when there is shortage, people do not have time to do everything they're supposed to do on patients, so they overlook management of pain."* (P4MG).

*"Lack of staff. We just fetch the patients and put them there on the bed and do not assess the patients thoroughly, we only check danger signs like bleeding and consciousness, pain is the last thing we think of, because there are things to be done, than managing pain, we prioritise"* (P10FO).

*". . . In this ward, we always have shortage because we nurse thirty plus patients in the ward. We do not give patient quality care, we concentrate on the routine"* (P7FG).

Other participants said that nurse-patient ratio contributed to the poor quality of nursing care where there is minimal staff on duty and thirty-two patients in the surgical wards. *". . . Remember like. . ., our ward admits 39 patients. According to patient-nurse ratio, there must be one nurse for six patients. And the shortage will lead to one nurse nursing eighteen patients. And due to shortage of nurses sometimes there is no one to manage patients pain"* (P1FO).

Another participant said: *"Okay.. . . The ratio of nurse-patient does not match because you find that there are two registered nurses, two enrolled nurses and two enrolled nursing assistants*

*to take thirty and more patients, by the way we are expected to do more other things than assessing and managing pain"* (P4MG).

A female participant for urology surgical ward indicated that: *"Shortage of staff, in particular nurses. According to me, because of the high nurse-patient ratio. We do not give enough time to patients post-operatively so that we can be able to assess pain, assist the patient and give the patient medication as it should be for pain, there are other duties, we do not just manage pain"* (P13FU).

Another participant mentioned that the unrealistic nurse-patients ratio in their ward creates a serious concern, especially when other nurses are on leave. She said: *"For example, in our ward three nurses. . ., are already on pension and they were never replaced. And on our leave profile, we allow two nursing categories to request leave, some get sick leave meaning, and already we are short staffed. People will be on leave and those who had resigned in the ward, are not replaced and we that get those who went on pension, are not replaced. We are overwhelmed and it is too much. And we end up not doing what we are supposed to do actually on patients, the workload is too much"* (P1FO).

Other participants spoke of nurses shortage in relation to nurses experience and seniority or rank. They felt that junior nurses are not adequately experienced to manage acute postoperative pain on their own. Therefore, as seniors they need to supervise them, while there are other duties at hand, which makes it difficult to manage acute post-operative pain. One participant said: *". . . Sometimes you find that there are only two sisters on duty neh. These juniors, normally they fetch the patients from theatre. They just fetch the patients from theatre and they do not even assess for pain. Sometimes they will inform you and you find that you are busy with other patients because it is the big ward"* (P10FO).

Based on the remarks provided by the participants, it appears that they are not purposefully mishandling the acute post-operative discomfort; rather, the heavy workload forces them to multitask by helping doctors with patients' ward rounds. They also prepare the patients for surgical procedure and to receive them from theatre after surgery.

**Inadequate skill competency to manage post-operative pain.** There are specific skills and expertise required in the surgical ward, so that the daily running is of high standard. The findings exposed the shortage of expertise required for the nurses in the surgical wards of the selected hospital to manage the acute post-operative pain. The participants stated that they are not adequately in-serviced to work with post surgical patients. Orthopaedic nurse mentioned this:

*"I can say lack of competency, or not given in-service training on how to manage patients in the unit. Remember like, if I am from another department not knowing that I must elevate the limb, we will keep on giving medication and patients complain of pain because the medication is not working; only to find that is because of swelling"* (P1FO).

Another participant said: *"Maybe I will say managing post-operative pain is not through medication, We need to know about the other methods, we should go for crush courses, I mean even long-term courses. Pointing to herself, like myself, the more junior you're, the more you need training, because you will not be knowledgeable in assessing and managing level of pain on patients"* (P4MG).

The other one said: *"Other barriers can be from lack of knowledge or incompetent in assessing level of pain and keeping a patient pain free, holding her chin with one hand, where you do not get enough of in-service training about the care of the post-operative patients"* (P9FO).

The findings also showed that the wards organise in-service training for the nursing personnel, however, due to busy schedules and shortage they usually do not attend. One participant said: *"The in-service training, they are having dates maybe. . . they do not suit us that day we are*

*so busy and we are missing the in-service training. So actually they supposed to do in-service every day because there is shortage of nurses, they can do spot teaching"* (P8FO).

**Interprofessional communication.**    Interprofessional communication emerged as a sub-theme under personnel related barriers. The participants revealed that the communication among themselves as surgical ward nurses and at times with the doctors contribute to unsatisfactorily post-surgical pain management. A participant said:

> *"And again, some nurses do not report, when a patient is in pain. If I'm in charge I look after all the patients and nurses, for me to know if there are any deviation I depend on my colleagues, I might miss other problems, if they are not communicated"* (P1FO).

It was also identified that some of the nurses are ignorant and do not document patient pain outcomes. *"Another way of communicating to others is through documentation, they are ignorant, some nurses do not document management provided to the patient, or the pain that patient reported"* (P3FG).

Some participants felt that doctors and nurses do not communicate effectively about the management of pain post surgery. A participant said: "Communication is a two-way process. A barrier arises if nurses fail to notify doctors when medicine or any other type of pain relief is ineffective" (P3FG).

Another nurse expressed that doctors are not communicating with them and it affects the management of pain post surgery. She said: "*For instead you find that the doctor did not write the whole information done on the patients in theatre, or did not prescribe strong pain medication to keep patient pain free, you're forced to phone the doctor, and he is busy operating another patient, it takes time to come and prescribe medication for the patient"* (P11FO).

## Theme 3. Patient related barriers

The findings further discovered that there are also post-operative pain management barriers due to patients issues. Participants stated that at times they fail to manage acute post-operative pain because some patients' cultures and religious beliefs do not allow them to verbalise or show that they are feeling pain. The participants said: *"Let me talk about cultural and religion, there are some people that will not verbalises that or give facial expressions that they have pain, but you will see it. If the nurses are not there to observe, like it can be another barrier that influence poor post-operative pain management"* (P1FO).

> *"I have noticed that, elderly patients do not report when feeling pain post surgery, maybe is due to cultural beliefs"* (P13FU).

Another participant said: *"They normally do not want to talk about the pain. They are having pain and they do not want to appear that they cannot handle pain, because they were oriented "man" not to cry. Some of these patients… because of age they are not well educated and there is a language barrier even when you want to assess pain there is misunderstanding between nurse and patient"* (P13FU).

Some participants expressed that they are unable to ask patients about post-operative pain because of language barriers. *"Sometimes we are unable to communicate with patients due to language barrier"* (P3FG).

It was also discovered that some of the prescribed pain medication you cannot administer when patient condition is not so well, for example low blood pressure. *If the blood pressure is low, we might also not give analgesics. We delay until blood pressure is within normal ranges"*
(P4MG).

Other participants mentioned that at times patients will ask for pain medication within short interval of another, and as nurses they do not administer it and it becomes a barrier. *"Most patients in our ward get spinal analgesia during surgery and it takes long to wean, like 6–8 hours. It becomes a challenge, when a patient complains of pain within an hour after surgery. You cannot give immediately because they are still under anaesthesia"* (P5FO).

## Discussion

The purpose of this article aimed to describe the barriers that hinder the surgical wards nurses to manage acute post-operative pain at the selected public hospital in Tshwane municipality, Gauteng Province, South Africa. Several barriers such as organisational/ management related, personnel related, and patient-related have been identified to hinder the surgical ward nurses at the selected hospital to manage acute post-operative pain. Similar barriers were previously identified in another settings [23].

The increased workload came out as one of the barriers hindering nurses to manage acute postoperative pain. Just like our study, some literature have related the increased workload to organisational or management functional structures [24, 25]. It should be noted that increased workload is one of the main issues the healthcare sector is facing [26]. The workloads of nurses in South Africa are greater than they have ever been, in which nurses and patients bear the consequences. Nurses perceive workload as important activities to be well planned if sound healthcare services are to be provided, especially in the surgical wards [27]. Complications and adverse events are more liable to occur among patients post surgery, hence the workload ratio of nursing personnel in these wards should be well planned.

The workload ratio is determined by a number of personnel against the number of patients. In our study shortage of nursing personnel in the surgical wards were highlighted as a barrier that results with unbalance, unrealistic and heavy workload. The relationship between nursing staffing levels and patient outcomes has been the main focus of recent research [27]. Moreover, strong evidence from earlier studies shows that high nursing workload at the unit level have a detrimental effect on patient outcomes [28, 29]. It should be noted that an unrealistic and heavy nursing workload can influence the nurses to perform certain important post-operative procedures over others like post-operative pain assessment and management. According to [28] it is because a nurse's workload has an impact on the amount of time she can devote to different tasks. Nurses who are overworked might not have enough time to complete tasks that directly affect patient safety. It is established that one of the main sources of job stress for nurses in a range of care environments, including surgical wards, is a heavy workload [4]. They are also prone to be demotivated, develop burnout, a high rate of absenteeism, a feeling of frustration and stress [26].

Nurses who are overworked, stressed and burned out may not be able to perform as well as they could, because their physical and mental resources may be depleted, and could have an impact on patient care and safety. In addition, [30] are of the opinion that patient care knowledge and execution errors may be caused by factors related to unrealistic and heavy workload, especially by nursing personnel who spend a lot of time with patients. A study conducted in India by [31] confirms that the minimum nurse-patient ratio in the surgical ward should be 1:5 to prevent nursing malpractice and extended hospital stays of patients in surgical wards. Some time heavy workload provides nurses with less time to keep patients pain free after the surgery. It is therefore recommended that to improve patient care and reduce the number of patients assigned to each nurse, the organisations need to add more nursing staff to the surgical wards.

The other personnel-related barrier to acute post-operative pain management was inadequate skill competency to manage post-operative pain. This is happening despite the acute postoperative pain management being widely recognised as important part of post surgery care, especially in developing countries [4]. This is primarily linked to health care providers, including nurses, being unaware of the issue or having insufficient knowledge [24, 32]. The degree of competency in assessing and managing pain following surgery is essential [33].

Nurses who lack competency, are not able to provide post-operative patients with quality care, especially assessment and management of pain. It is therefore, important for the surgical ward nurses to be equipped with the knowledge and abilities needed to assess, diagnose, plan, intervene, and evaluate the results of post-operative pain.

Our findings also showed that a lack of interprofessional communication between nurses and other health care providers it is a barrier to assess and manage post-operative pain. Communication among health care providers or patients, is a key element in providing high quality health care, which leads to patient satisfaction. According to [34] inadequate communication starts before the surgery, in which the type of anaesthesia is not communicated to all the perioperative team including the surgical ward nurses. Inadequate communication, especially doctors orders regarding medication to keep post-operative patients free from pain makes it difficult for the nurses to render quality nursing care [35]. [36] confirm that the procedures not documented by doctors in patients files, has a negative impact on patient care.

Patient related barriers also emerged as a main theme that affect the assessment and management of acute post-operative pain. The discussion was based on culture and religious beliefs, language barrier and patient physical condition. The discussion was unlike in the study conducted in Lebanon, which described patient related pain assessment and management barriers to fear of addiction, side effects, receiving more injections and additional costs [37]. Drawing from the findings of the current study it was evident that most of the post-operative patients are not reporting pain due to some own beliefs. This finding share same sentiments with the study that investigated the effects of a nurse-led pain management programme on pain intensity after thoracic surgery in Nigeria, in which patients instead of reporting pain when it occurs, prefer to bear it, modify their posture or employ different non-pharmacological coping mechanisms [18, 38].

The participants of the study conducted on patients satisfaction and related barriers of pain management in hospitals believed that good people should avoid talking about pain and that pain medication should only be administered to those who experience severe pain [37]. It is obvious that notwithstanding advancements in pain medicine and pain teaching techniques, effective pain management is proportionate to both clinical and cultural preferences [38]. A study conducted in Ethiopia by [4] highlighted that healthcare providers, including nurses should increase their knowledge of different cultures, such as learning others languages to improve nurse-patient communication. Some authors argue that it is important to take into account the cultural validity of self-reported pain assessment instruments and make necessary modifications to accommodate cultural perspectives and customize language for the local population [39].

Given that meeting each person's unique health needs always begins with communication, language plays a crucial role in the healthcare communication process [40]. Some of the participants in our study reported that it is difficult to assess and manage acute post-operative pain because of language barriers. This finding is not surprising because language barriers prevent a large portion of South African population from accessing healthcare [40]. As a results, in the immediate postoperative setting, patients who do not speak English may be at risk of receiving insufficient pain management [41]. [40] also argue that language barriers impede

communication, and can lead to misinterpretations, incorrect diagnoses, inadequate or insufficient patient assessment and delayed management.

Language barriers, might compromise patient pain threshold and increase provider bias in pain management. Therefore, it is recommended that effective communication using patients language has therapeutic benefits in nurse-patient interactions. Patient physical condition was also raised as a barrier for a proper acute post-operative pain assessment and management. Previous literature revealed that patients on opioid treatment for chronic pain pose a challenge because their daily baseline dosages of postoperative analgesic are usually insufficient [42, 43]. [44] added that psychological factors such as depression, anxiety and other mental health issues can increase risk for high levels of postoperative pain, and need to be assessed and managed differently. Since every patient experiences pain differently, there is a need for numerous options for how to treat it [45].

## Conclusion

This study reported on the barriers that hinder the surgical wards nurses to manage acute post-operative pain at the selected public hospital in Tshwane municipality, Gauteng Province, South Africa. The barriers were based on organisational, personnel and patients. It became evident in the findings that most participants revealed that the increased workload and nurse-to-patient ratio in the surgical wards compelled them not to assess post-operative pain and manage patients according to their individual needs. The managers are aware of the increased workload in the surgical wards; they should therefore develop a nurse-patient ratio policy to prevent participants' dissatisfaction and burnout. It is the responsibility of surgical ward managers to book participants for overtime to reduce workload and thus support their participants.

## Implications

The development, implementation and education of acute post-operative pain management policies and guidelines by surgical ward nurses and managers can improve the quality of nursing care provided to the patients post surgery at the selected public hospital in Tshwane municipality, Gauteng Province.

## Limitations

The findings obtained in this study were collected from participants who worked in adult general, urology, and orthopaedic surgical wards of a selected academic public hospital in Gauteng Province. Participants from other hospitals in Gauteng Province would have provided a wider information in this regard.

## Acknowledgments

The authors would like to acknowledge the professional nurses at the selected surgical wards who participated in this study and shared their views on barriers hindering post-operative pain level assessment.

## Author Contributions

**Supervision:** Melitah Molatelo Rasweswe, Ramadimetja Shirley Mooa.

**Writing – original draft:** Nnene Melia Makou.

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
