## [Decision Letter · Decision Letter 0]

13 Aug 2024

PONE-D-24-21430Voices of surgical wards nurses on barriers hindering acute post-operative pain management at Tshwane municipality, South AfricaPLOS ONE

Dear Dr. Melia,

Thank you for submitting your manuscript to PLOS ONE. After careful consideration, we feel that it has merit but does not fully meet PLOS ONE’s publication criteria as it currently stands. Therefore, we invite you to submit a revised version of the manuscript that addresses the points raised during the review process.

We look forward to receiving your revised manuscript.

Kind regards,

Sunil Shrestha

Academic Editor

PLOS ONE

Journal Requirements:

2. Please update your submission to use the PLOS LaTeX template. The template and more information on our requirements for LaTeX submissions can be found at http://journals.plos.org/plosone/s/latex."

3. Please amend the manuscript submission data (via Edit Submission) to include authors Dr. Rasweswe Melitah Molatelo and

Dr. Mooa Ramadimetja Shirley.

Additional Editor Comments:

Major Revision

Reviewers' comments:

Reviewer's Responses to Questions

**Comments to the Author**

1. Is the manuscript technically sound, and do the data support the conclusions?

Reviewer #1: Yes

2. Has the statistical analysis been performed appropriately and rigorously? 

Reviewer #1: N/A

3. Have the authors made all data underlying the findings in their manuscript fully available?

Reviewer #1: Yes

4. Is the manuscript presented in an intelligible fashion and written in standard English?

Reviewer #1: Yes

5. Review Comments to the Author

Reviewer #1: Recommendation

Minor Revision

Thank you for sharing your manuscript titled: Voices of surgical wards nurses on barriers hindering acute post-operative pain management at Tshwane municipality, South Africa. An interesting study, a good justification is needed in the ethical consideration section about informed consent.

Below are further feedback for consideration:

Objective

I will suggest that you rephrase the objective and use such term as “uncovering”….. instead of “Identifying”

Uncovering barriers that hinder the surgical wards nurses to manage acute postoperative pain at the selected public hospital in Tshwane municipality, Gauteng Province, South Africa.

Population and sampling

What are the reasons why other categories are excluded from the study? Are other categories not eligible and why.

You mention professional nurses who are temporary or on contract are not included? Justify why they are excluded?

Could you please rephrase the sentence? (see Manuscript)

Ethical considerations

Was the informed consent verbal or written?

If you use AI such as ChatGPT, Please acknowledge it.

6. PLOS authors have the option to publish the peer review history of their article (what does this mean?). If published, this will include your full peer review and any attached files.

Reviewer #1: No

---

## [Author Response · Author response to Decision Letter 0]

30 Aug 2024

PONE-D-24-21430

Voices of surgical wards nurses on barriers hindering acute post-operative pain management at Tshwane municipality, South Africa

PLOS ONE

Dear Dr. Melia,

Thank you for submitting your manuscript to PLOS ONE. After careful consideration, we feel that it has merit but does not fully meet PLOS ONE’s publication criteria as it currently stands. Therefore, we invite you to submit a revised version of the manuscript that addresses the points raised during the review process.

Author response: Included.

Author response: Included.

Author response: Included.

If applicable, we recommend that you deposit your laboratory protocols in protocols.io to enhance the reproducibility of your results. Protocols.io assigns your protocol its own identifier (DOI) so that it can be cited independently in the future. 

For Instructions see: https://journals.plos.org/plosone/s/submission-guidelines#loc-laboratory-protocols. Additionally, PLOS ONE offers an option for publishing peer-reviewed Lab Protocol articles, which describe protocols hosted on protocols.io. 

Read more information on sharing protocols at https://plos.org/protocols?utm medium=editorial-email&utm source=authorletters&utm campaign=protocols. 

We look forward to receiving your revised manuscript.

Kind regards,

Sunil Shrestha

Academic Editor

PLOS ONE

Journal Requirements:

 Author response: We have followed the above-mentioned PLOS ONE style templates.

2. Please update your submission to use the PLOS LaTeX template. The template and more information on our requirements for LaTeX submissions can be found at http://journals.plos.org/plosone/s/latex."

 Author response: We have followed the PLOS One template available at https://journals.plos.org/plosone/s/latex.

3. Please amend the manuscript submission data (via Edit Submission) to include authors Dr. Rasweswe Melitah Molatelo and Dr. Mooa Ramadimetja Shirley.

 Author response: We have included the above-mentioned authors.

Reviewers' comments:

Reviewer's Responses to Questions

Comments to the Author

1. Is the manuscript technically sound, and do the data support the conclusions?

Reviewer #1: Yes

Author response: We appreciate the reviewer’s feedback, Thank you!

2. Has the statistical analysis been performed appropriately and rigorously?

Reviewer #1: N/A

Author response: Thank you for pointing this out. The reviewer is correct, as our manuscript is based on the qualitative research approach. 

3. Have the authors made all data underlying the findings in their manuscript fully available?

Reviewer #1: Yes

Author response: We appreciate the reviewer’s feedback, Thank you!

4. Is the manuscript presented in an intelligible fashion and written in standard English?

Reviewer #1: Yes

Author response: We appreciate the reviewer’s feedback, Thank you!

5. Review Comments to the Author

Reviewer #1: Recommendation

Minor Revision

Thank you for sharing your manuscript titled: Voices of surgical wards nurses on barriers hindering acute post-operative pain management at Tshwane municipality, South Africa. An interesting study, a good justification is needed in the ethical consideration section about informed consent.

Author response: We appreciate the reviewer positive encouraging comment.

Below are further feedback for consideration:

Objective:

I will suggest that you rephrase the objective and use such term as “uncovering”….. instead of “Identifying”

Uncovering barriers that hinder the surgical wards nurses to manage acute postoperative pain at the selected public hospital in Tshwane municipality, Gauteng Province, South Africa.

Author response: As suggested by the reviewer, we have rephrased the objective and the term “uncovering” was used and the changes are highlighted with colour ‘blue’ in line 33, page 2 of the 'Revised Manuscript with Track Changes'. 

Population and sampling:

What are the reasons why other categories are excluded from the study? Are other categories not eligible and why.

Author response: We appreciate and respect the reviewer’s feedback. Other nursing categories we excluded because the registered professional nurses, registered by a nursing regulatory authority South African Nursing Council are have more knowledge, able to provide supervision and, be responsible and accountable for the delegated nursing activities to the subordinates during the rendering of quality nursing care to the post-operative patients. All other categories are eligible to render quality nursing care to the post-operative patients as they have undergone education and training in the accredited universities and nursing education institutions. However, in this study we were focusing on the registered professional nurses who are able to provide supervision, able to implement, monitor and evaluate post-operative patients and as well as to develop, implement, monitor and evaluate the in-service education program regarding care of the post-operative patients to increase satisfaction and reduce long of hospital stays. Sentence not changed, highlighted with colour ‘blue’ in line 149, page 7 of the 'Revised Manuscript with Track Changes'. 

You mention professional nurses who are temporary or on contract are not included? Justify why they are excluded?

Author response: Thank you for pointing this out. We believed that the professional nurses who are on temporary or contract work do not have a permanent job in the public hospital's surgical wards. In this study, we were focusing on the professional nurses, registered by a nursing regulatory authority South African Nursing Council working in the surgical wards of the selected public hospital and have more than twelve months working experience. Sentence rephrased and highlighted with colour ‘blue’ in line 150, page 7 of the 'Revised Manuscript with Track Changes'.

Could you please rephrase the sentence? (see Manuscript)

Author response: We appreciate the reviewer’s feedback. We have added the rephrased sentence and highlighted with colour ‘blue’ in line 150, page 7 of the 'Revised Manuscript with Track Changes' as suggested by the reviewer. 

Sentence was: Professional nurses who were on temporary or contract work were also excluded.

Rephrased sentence: In addition, participants in this study were not registered professional nurses employed on a contract or temporary basis in the public hospital's surgical wards.

Ethical considerations:

Was the informed consent verbal or written?

Author response: We appreciate the reviewer’s feedback. A written informed consent was obtained prior to participation. Once the participant agreed to take part in the study were given a written informed consent, signed it to avoid coercion. Sentence not changed, highlighted with colour ‘blue’ in line 162, page 7 of the 'Revised Manuscript with Track Changes'. 

If you use AI such as ChatGPT, Please acknowledge it.

Author response: We would like to appreciate QuillBot AI, for assisting with rephrasing of sentences.

6. PLOS authors have the option to publish the peer review history of their article (what does this mean?). If published, this will include your full peer review and any attached files.

Do you want your identity to be public for this peer review? For information about this choice, including consent withdrawal, please see our Privacy Policy.

Reviewer #1: No

---

## [Decision Letter · Decision Letter 1]

18 Dec 2024

Voices of surgical wards nurses on barriers hindering acute post-operative pain management at Tshwane municipality, South Africa

PONE-D-24-21430R1

Dear Dr. Melia,

We’re pleased to inform you that your manuscript has been judged scientifically suitable for publication and will be formally accepted for publication once it meets all outstanding technical requirements.

Kind regards,

Vanessa Carels

Staff Editor

PLOS ONE

Additional Editor Comments (optional):

Reviewers' comments:

Reviewer's Responses to Questions

**Comments to the Author**

1. If the authors have adequately addressed your comments raised in a previous round of review and you feel that this manuscript is now acceptable for publication, you may indicate that here to bypass the “Comments to the Author” section, enter your conflict of interest statement in the “Confidential to Editor” section, and submit your "Accept" recommendation.

Reviewer #1: All comments have been addressed

2. Is the manuscript technically sound, and do the data support the conclusions?

Reviewer #1: Yes

3. Has the statistical analysis been performed appropriately and rigorously? 

Reviewer #1: Yes

4. Have the authors made all data underlying the findings in their manuscript fully available?

Reviewer #1: Yes

5. Is the manuscript presented in an intelligible fashion and written in standard English?

Reviewer #1: Yes

6. Review Comments to the Author

Reviewer #1: The authors have attended to my comments and I am satisfied with their comments.

The editor should accept the paper if they meet with other edotrial requirements.

Thank you.

7. PLOS authors have the option to publish the peer review history of their article (what does this mean?). If published, this will include your full peer review and any attached files.

Reviewer #1: No

---

## [Editor Report · Acceptance letter]

31 Dec 2024

PONE-D-24-21430R1 

PLOS ONE

Dear Dr. Makou, 

I'm pleased to inform you that your manuscript has been deemed suitable for publication in PLOS ONE. Congratulations! Your manuscript is now being handed over to our production team.

Kind regards, 

on behalf of

Dr. R&R PLOS 

Staff Editor

PLOS ONE